# DEVBENCH: A multimodal developmental benchmark for language learning

**Alvin W. M. Tan, Sunny Yu,**
**Bria Long, Wanjing Anya Ma, Tonya Murray,**
**Rebecca D. Silverman, Jason D. Yeatman, Michael C. Frank**
Stanford University
{tanawm, syu03, bria, wanjingm, tonyamur,
rdsilver, jyeatman, mcfrank}@stanford.edu

## Abstract

How (dis)similar are the learning trajectories of vision–language models and children? Recent modeling work has attempted to understand the gap between models' and humans' data efficiency by constructing models trained on less data, especially multimodal naturalistic data. However, such models are often evaluated on adult-level benchmarks, with limited breadth in language abilities tested, and without direct comparison to behavioral data. We introduce DEVBENCH, a multimodal benchmark comprising seven language evaluation tasks spanning the domains of lexical, syntactic, and semantic ability, with behavioral data from both children and adults. We evaluate a set of vision–language models on these tasks, comparing models and humans on their response patterns, not their absolute performance. Across tasks, models exhibit variation in their closeness to human response patterns, and models that perform better on a task also more closely resemble human behavioral responses. We also examine the developmental trajectory of OpenCLIP over training, finding that greater training results in closer approximations to adult response patterns. DEVBENCH thus provides a benchmark for comparing models to human language development. These comparisons highlight ways in which model and human language learning processes diverge, providing insight into entry points for improving language models.

## 1 Introduction

Humans are remarkably good language learners, acquiring language rapidly in the first few years of life without much explicit supervision. Recent machine learning models are also able to acquire some aspects of human language, although they require much larger quantities of language input than a typical human would receive [1, 2]. This "data gap" reflects differences in the learning processes and mechanisms employed by human language learners and language models; understanding the nature of these differences is pivotal for the joint goals of building better cognitive models of language learning and building more data-efficient language models.

Recent work has attempted to bridge this data gap by constructing models that are trained on less data, including on naturalistic data from children [2–5]. However, these models have typically been evaluated using benchmarks that reflect adult human performance, whether explicitly (i.e., with comparison data collected from adult participants) or implicitly (i.e., high accuracy on some recognition task). Such evaluation methods are not appropriate to the goal of understanding whether developmentally realistic data leads to human-like learning. We would not expect child performance to be similar to adult performance on these tasks for various reasons related to both language competence (i.e., children's vocabulary knowledge) and to their co-developing cognitive abilities

38th Conference on Neural Information Processing Systems (NeurIPS 2024) Track on Datasets and Benchmarks.

(e.g., working memory limitations). Instead, it is crucial to evaluate models on benchmarks that can indicate whether the language ability gained by machine learning models matches the language ability gained by children when exposed to similar developmental data.

Research in child language acquisition has observed that children learn different aspects of language at different rates, and the acquisition of different levels of representation also interact in complex and nonlinear ways [6–10]. Thus, in order to characterise models' language learning performance, we should evaluate multiple levels of linguistic representation, including the lexicon, syntax, and semantics – ideally how these correspond to children's development at different ages.

Additionally, to conduct methodologically rigorous cognitive evaluation of machine learning models, it is important to compare human and model performance directly (rather than assuming that humans will perform well), and to ensure that evaluation conditions are as similar as possible for models and humans [11–13]. Notably, many language evaluations with infants and toddlers are multimodal in nature, because this method permits the measurement of language comprehension without the additional working memory and motor control demands of language production. Young children can use nonverbal response methods such as looking or pointing, which can nonetheless reflect their language abilities. The incorporation of multimodality has an additional advantage of introducing grounding, which has been suggested as a possible solution to the data gap [1].

These observations provide a set of desiderata for a developmentally appropriate evaluation of language models:

1. Wide dynamic range of difficulty
2. Multiple levels of linguistic representations
3. Corresponding data from children
4. High similarity in evaluation method between models and humans

To create a benchmark that satisfies these desiderata, we introduce **DEVBENCH**, a multimodal benchmark of language evaluation tasks. DEVBENCH includes a suite of seven tasks measuring lexical, syntactic, and semantic ability, with human data from both children and adults. Notably, the primary evaluation metric reflects models' similarity to human response patterns, rather than absolute performance levels, allowing us to capture finer-grained details about human–model similarity.

We evaluate a set of vision–language models on DEVBENCH, including not only state-of-the-art models but also smaller models and models trained on developmentally realistic data. We additionally investigate how DEVBENCH performance varies over model training by evaluating intermediate checkpoints of OpenCLIP [14]. Evaluation results suggest that current vision–language models exhibit variation in their human-likeness, suggesting further areas of research to develop language models that more closely approximate human language learning.

## 2 Related work

### 2.1 Multimodal benchmarks

Since the advent of multimodal models, various benchmarks have been developed to evaluate their performance [15]. A majority of these benchmarks primarily involve visual question answering [16–21]; other tasks include image captioning [18, 22], prediction [23], and retrieval [17, 22].

Most tasks in these multimodal benchmarks probe models' perception, reasoning, and knowledge, evaluating models on domains including action prediction, counting, relational reasoning, or optical character recognition. Correct responses to these tests require the conjunction of many skills – notably, all reasoning tasks also require perception, and perception tasks in turn broadly require object knowledge. Furthermore, most multimodal benchmarks focus on models' visual understanding, and no existing benchmark focuses specifically on the linguistic abilities of multimodal models.

### 2.2 Developmentally inspired evaluation

Some recent work in machine learning evaluation has used benchmarks inspired by developmental psychology and cognitive science. For example, the Large Language Model Response Score (LRS) [24] draws from key experiments in the child development literature to construct question answering tasks for large language models, though it does not involve comparison to data from

Table 1: Characteristics of multimodal and developmentally inspired benchmarks.

| Benchmark | Evaluation features | | | | Domains | | | Human data | |
|---|---|---|---|---|---|---|---|---|---|
| | Multi-modal | Zero-shot | Nonverbal response | Develop-mental | Lexicon | Syntax | Semantics | Children | Adult |
| LAMM [18] | ✓ | ✓ | | | | | | | |
| MultiBench [23] | ✓ | ✓ | | | | | | | |
| GEM [22] | ✓ | ✓ | ✓ | | | | | | |
| MMBench [20] | ✓ | ✓ | ✓ | | | | | | |
| SEED-Bench [28] | ✓ | ✓ | ✓ | | ✓ | | ✓ | | |
| MME [19] | ✓ | ✓ | ✓ | | ✓ | | ✓ | | |
| Perception Test [21] | ✓ | ✓ | ✓ | | ✓ | | ✓ | | |
| M3Exam [29] | ✓ | ✓ | | ✓ | | | | | ✓ |
| LRS [24] | | ✓ | | ✓ | | | | | |
| InfLevel [25] | | ✓ | ✓ | ✓ | | | | | |
| Zorro [3] | | ✓ | ✓ | ✓ | | ✓ | | | |
| MEWL [26] | ✓ | | ✓ | ✓ | ✓ | | ✓ | | ✓ |
| ModelVsBaby [27] | ✓ | ✓ | ✓ | ✓ | ✓ | | | ✓ | |
| DEVBENCH | ✓ | ✓ | ✓ | ✓ | ✓ | ✓ | ✓ | ✓ | ✓ |

children. In addition, most of its tasks were originally multimodal (with visual stimuli and verbal prompts), and it is not clear how the translation into a unimodal verbal task would affect human performance. In the visual domain, the Infant-Level Physical Reasoning Benchmark (InfLevel) [25] aimed to evaluate video models on physical reasoning, drawing from classic violation of expectation tasks in the infant cognition literature related to the principles of continuity, solidity, and gravity. However, this benchmark also did not have a direct comparison with human data. The Machine Word Learning (MEWL) benchmark [26] is a multimodal evaluation that builds upon hypothesised mechanisms of few-shot word learning in children, and requires the inference of the meaning of novel words from a set of visual scenes with referentially ambiguous labels. The benchmark contains data from adults, but their poor performance on some subsets of these trials (e.g., those requiring pragmatic implicature) suggests that children might find these tasks very difficult. More recently, the ModelVsBaby benchmark [27] has assessed out of distribution object recognition with corresponding data from 2-year-olds, providing a first step towards direct developmental model–human comparison.

Other work assessing developmental models has also developed ad-hoc evaluations. The most common method has been to design evaluation sets that resemble other machine learning model tasks (e.g., grammatical acceptability, image classification) while restricting the domain to a developmentally relevant domain (i.e., only including vocabulary familiar to the model, or choosing image categories that are child-relevant). This method has been applied both to language models (e.g., the Zorro benchmark on BabyBERTa models [3]), and to multimodal models (e.g., the Konkle Objects evaluation on CVCL models [4]). However, again in these cases there are typically no child data, and the assumption that children would perform well in such evaluations is not directly evaluated. We summarise the characteristics of a range of multimodal and developmental benchmarks in Table 1.

### 2.3 Model learning trajectory analyses

A few studies have used developmental approaches to analyse model learning trajectories. Chang and Bergen [30] examined the age of acquisition of different words by measuring the change in mean surprisal of a word over training epochs, finding dissimilarities in the predictors of age of acquisition in models and children. Evanson et al. [31] examined the age of acquisition of different syntactic structures by measuring the point at which models began preferring the grammatical sentence in a grammatical acceptability task. We extend these approaches here by examining training trajectories across multiple levels of linguistic representation.

## 3 DEVBENCH description

DEVBENCH contains seven tasks across lexical, syntactic, and semantic domains. Each task is accompanied by item-level human data so that full human response distributions can be compared to model scores. The lexical tasks measure vocabulary knowledge, operationalised as the ability to correctly pick out the visual referent of a noun label. The syntactic tasks measure grammatical knowledge, operationalised as the ability to correctly pick out a scene containing the correct relations

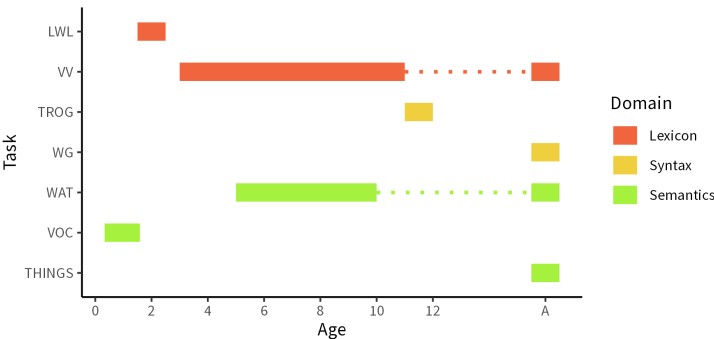

Figure 1: Tasks in DEVBENCH arranged by linguistic domain, along with the ages for which corresponding human data are available. A: Adult.

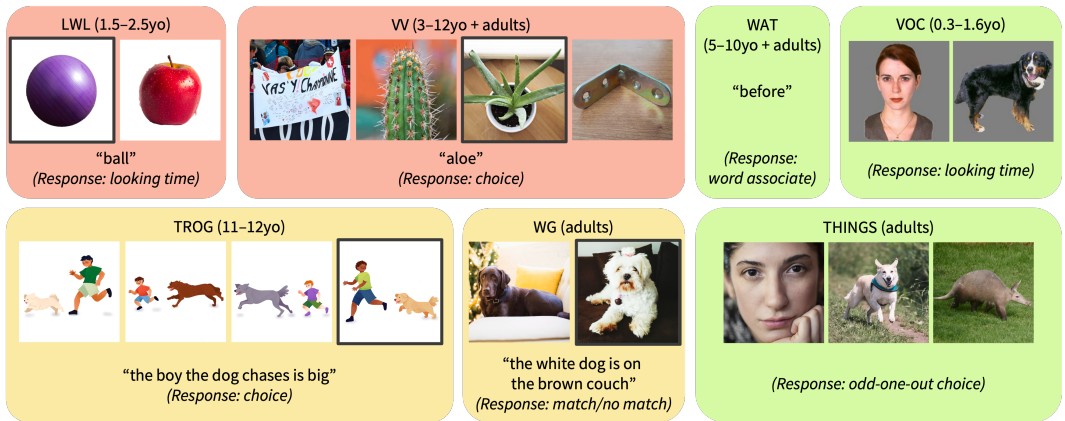

Figure 2: Sample trials for each task in DEVBENCH.

among its constituent members when given a sentential label. The semantic tasks measure conceptual (visual or linguistic) representation space via representational similarity. A schematic of the tasks and the ages of participants contributing data for each task is shown in Figure 1, and sample trials are shown in Figure 2. All code and data are available at `github.com/alvinwmtan/dev-bench`; all data are licensed under CC BY-NC-SA or more permissible licences, and have been anonymised to eliminate any personally identifiable information.

## 3.1 Tasks

**Lexicon: Looking-while-listening (LWL)**    In this task, children are presented with two visual images (one target and one distractor) as well as a verbal cue (e.g., "Look at the dog!") [32]. Children's proportion looking time to each image is measured. We used data from three datasets to improve age coverage: 18-month-old data from Adams et al. [33], 24-month-old data from Frank et al. [34], and 30-month-old data from Donnelly and Kidd [35] (total $N = 294$). Some images in the original stimuli set were not shareable due to license restrictions; in these cases, we used replacement stimuli matched for visual and semantic properties.

**Lexicon: Visual vocabulary task (VV)**    In this task, participants hear a word (e.g., "swordfish") and then see four visual images corresponding to the target and three distractor images that range in similarity to the target word (close, far, and distal) [36]. Participants respond by choosing the image that they think matches the verbal cue. We used an unpublished dataset from Long et al. [36] containing responses from 3- to 12-year-olds as well as adults (total $N = 1780$).

**Syntax: Test of Receptive Grammar (TROG)**    In this task, participants are presented with four visual images (one target and three distractors) and then hear a phrase or sentence cue (e.g., "The

brown horse chases the white dog") [37]. In most trials, the distractors are constructed such that they would align with a label that contains the same words as the cue but in a different order (e.g., an image containing a brown dog chasing a white horse). Participants respond by choosing the image that they think matches the verbal cue. We used stimuli and unpublished data from Silverman and Yeatman [38], containing responses from children aged 11–12 (total $N = 514$).

**Syntax: Winoground-NoTag (WG)**   This task contains trials with two images and two sentence labels, such that each image has one corresponding label, and the two labels only differ in word order [39]. Human ratings were collected by asking adult crowdworkers whether one particular label matched one particular image, and an image–caption score was calculated as the proportion of affirmative responses (total $N = 171$). For comparability to the other tasks in this benchmark, we considered the image–caption scores for the two images and one sentence of each trial, converted to proportions. We included only trials labelled "NoTag" from [40], which reflect vanilla Winoground trials that rely more narrowly on syntax (as opposed to also requiring pragmatics or other language abilities).

**Semantics: Free word association task (WAT)**   In this task, participants were presented with a cue word, and asked to name the first word association that came to mind. We use association frequencies as a proxy for similarity scores. We combined a child dataset from Entwisle [41] with a subset of the adult dataset from Nelson et al. [42] comprising all cue words found in the child dataset (total $N > 7040$), and thresholded the included responses to omit idiosyncratic responses.

**Semantics: Visual object categorization (VOC)**   In this task, 4-, 10-, and 14-month-old infants saw pairs of images, and proportion looking times to each image was measured [43] (total $N = 73$). We used replacement stimuli for some images that were not shareable in the original stimuli set. Dissimilarities between images were calculated as the difference in proportion looking times to the two images.

**Semantics: THINGS similarity ratings**   In this task, adult participants did a triplet odd-one-out task one triads constructed from the THINGS database [44, 45] (total $N = 12340$). These were used to generate a sparse positive similarity embedding [46], which we used to calculate pairwise similarities among images.

### 3.2   Human–model comparison

Because we were interested in response patterns, we conducted human–model comparison by examining the (dis)similarities in human and model distributions in responses.

Our goal on the lexicon and syntax tasks was to compare the distribution of human choices across response options to model choices. We obtained image–text matching logits for each response option, and calculated the *softmax-optimised Kullback–Leibler divergence* of human responses from model responses. This novel metric was operationalised as the minimum KL divergence between the human response probability distributions $\mathbf{h}$ from model logits $\mathbf{m}$, optimizing the softmax exponent $\beta$. We average this divergence across trials $t$, and calculate each distribution across images $i$:

$$D_{\text{KL}}^{*}(h \parallel m) = \min_{\beta} \frac{1}{t} \sum_{t} D_{\text{KL}} \left( \mathbf{h}_t \parallel \frac{e^{\beta \mathbf{m}_{it}}}{\sum_i e^{\beta \mathbf{m}_{it}}} \right)$$

For WAT, we calculated the softmax-optimised KL divergence of human association probabilities from softmaxed model text embedding similarities, averaged over all trials. For the visual semantic tasks, we instead conducted human–model comparison by applying representational similarity analysis (RSA) [47] on human and model representational similarity matrices, which represents correlations in the representational geometries of humans and models. All comparisons were conducted within each age bin when applicable.

Our method applies to models from which image–text matching scores could be directly extracted (i.e., similarity models). More recent models have alternatively integrated visual and language inputs via conditional text generation (i.e., generation models). There is as yet limited consensus for the best method to obtain image–text matching scores for generation models; we conducted two exploratory

Table 2: Model characteristics and performance across all tasks, demonstrating variation across models. Arrows indicate the direction of better performance (i.e., lower is better vs. higher is better). Bolded results indicate most human-like result on a task.

| Model | # params | # images | Lexicon | | Syntax | | Semantics | | |
|---|---|---|---|---|---|---|---|---|---|
| | | | LWL ($\downarrow$) | VV ($\downarrow$) | TROG ($\downarrow$) | WG ($\downarrow$) | WAT ($\downarrow$) | VOC ($\uparrow$) | THINGS ($\uparrow$) |
| CLIP-base [48] | 149M | 400M | 0.014 | 0.205 | 0.732 | 0.256 | 0.495 | -0.081 | **0.397** |
| CLIP-large [48] | 428M | 400M | 0.013 | **0.179** | 0.692 | 0.256 | 0.495 | 0.005 | 0.246 |
| ViLT [49] | 87M | 4.1M | 0.009 | 0.326 | 0.682 | 0.252 | 0.495 | -0.053 | 0.127 |
| FLAVA [50] | 350M | 70M | 0.013 | 0.197 | 0.912 | 0.254 | 0.495 | -0.042 | 0.189 |
| BLIP [51] | 252M | 14M | 0.010 | 0.193 | **0.576** | 0.259 | 0.495 | -0.104 | 0.185 |
| BridgeTower [52] | 333M | 4M | **0.008** | 0.265 | 0.584 | **0.250** | 0.495 | -0.095 | 0.345 |
| OpenCLIP-H [53] | 1.0B | 32B | 0.012 | 0.188 | 0.683 | 0.255 | **0.495** | 0.031 | 0.227 |
| SigLIP [54] | 800M | 9B | 0.067 | 0.612 | 0.888 | 0.258 | 0.495 | -0.028 | 0.192 |
| CVCL [4] | 26M | 600K | 0.060 | 0.740 | 0.911 | 0.258 | 0.495 | **0.138** | 0.175 |
| Human | | | 0.010 | 0.091 | 0.028 | | | 0.251 | |
| Random (OpenCLIP) | 1.0B | 0 | 0.087 | 0.740 | 0.908 | 0.258 | 0.495 | 0.246 | 0.054 |

evaluation methods relying on next-token prediction and on log-likelihood measurement. More details on the evaluation of generation models can be found in Appendix C.

## 3.3 Baselines

We also constructed two baselines for the benchmark to demonstrate the dynamic range that is possible for each task. First, we calculated a human baseline for tasks on which participant-level data were available (LWL, VV, TROG, VOC). To estimate this baseline, we randomly split the participants into two groups and calculated the between-group softmax-optimised KL divergence or RSA similarity as appropriate, repeating for 1000 random splits. We used the median result as a point estimate of the human baseline, which serves as a positive baseline for our tasks. Note that this baseline is an underestimate of the true values, because the results are not corrected upwards for the attenuation due to splitting the data in half.

We also added a random baseline for all tasks, generated using a random initialisation of the OpenCLIP model. The random baseline serves as a negative baseline for our tasks.

## 4 Benchmark

We evaluated a diverse set of vision–language models on our benchmark, using an NVIDIA T4 GPU, an NVIDIA A40 GPU, or CPUs (depending on resource availability). A summary of model performance for each task (averaged across all ages) is shown in Table 2, along with model characteristics (number of parameters and size of training set). For lexicon and syntax tasks as well as WAT, we report $D_{\text{KL}}^*$, for which *lower* scores indicate greater human-likeness in response patterns. For visual semantic tasks (VOC, THINGS), we report RSA similarity, for which *higher* scores indicate greater human-likeness in response patterns. Divergences are not comparable across datasets due to different features of each dataset. A description of all models and more details on the evaluation setup can be found in Appendix A.

Overall, CLIP-large and OpenCLIP-H performed relatively well on the lexicon and syntax tasks. CVCL, which was trained on a small set of annotated head-mounted camera from infants [4], was mostly dissimilar to humans in lexicon and syntax tasks, even on the looking-while-listening (LWL) task, which was administered to young children. However, CVCL outperformed other models on the visual object categorization task, suggesting that its visual representational space was more similar to infants'. SigLIP was also unusually poor-performing (especially given its general performance and accuracy), potentially suggesting that aspects of its training (e.g., its objective function) may have resulted in non-alignment with humans. We thus excluded it as an outlier from further analyses.

## 5 Analysis

To further understand the relationship between model and human responses, we conduct more fine-grained analyses, aiming to answer three specific research questions:

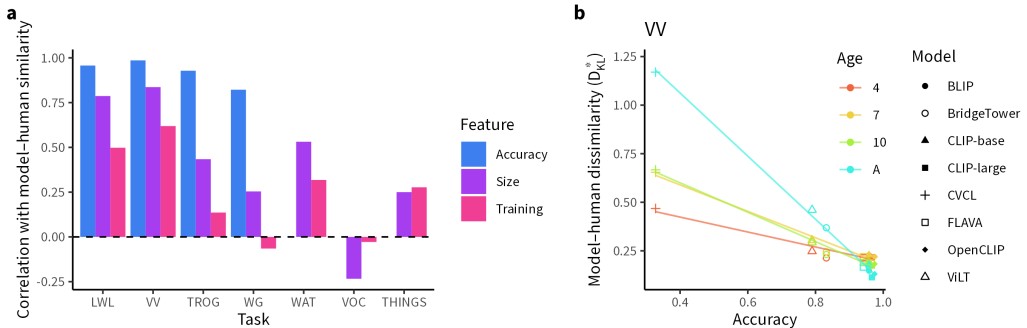

Figure 3: (a) Correlations between model–human similarity and task accuracy, log number of parameters (size), and log number of training images for each task (training), averaged across ages. Accuracy correlates the most strongly with model–human similarity, followed by size, then training. (b) Model–human dissimilarity as a function of task accuracy for each model on the Visual Vocabulary task. Higher performing models showed a closer correspondence to behavioral patterns from children and adults. A: Adult.

1. How does model–human similarity relate to other features of the models, namely their size (in terms of parameters), their training dataset size, and their overall accuracy on the tasks?
2. How does model–human similarity change over the course of model training, and in particular can we elucidate "developmental" trends in model training?
3. On which items are models and humans most dissimilar? On which items are they most similar?

## 5.1  Model feature analysis

To understand the variation in human-likeness exhibited by different models, we considered a range of model features that may affect response patterns, namely the number of parameters of the model, the number of examples in its training set, and its accuracy on the task (for lexicon and syntax tasks, which have a "correct" answer). For each task and each feature, we calculated Pearson's correlations between feature values and model–human similarities; number of parameters and number of training images were log-transformed. Similarity–feature correlations are shown in Figure 3a.

Overall, we found that model–human similarity was most correlated with task accuracy, with consistently high correlations across all ages and tasks. Model size also correlates relatively well with model–human similarity for most tasks except for VOC, which is reasonable given that VOC was conducted on the youngest participants (aged 4–14 months). In contrast, the number of training examples exhibited the poorest correlations with model–human similarity, suggesting that dataset size may not be as informative about human-likeness as accuracy or model size.

To illustrate the relationship between accuracy and model–human similarity, we plot these values for the Visual Vocabulary (VV) task in Figure 3b, which is also the task for which we have the largest coverage across age groups. In the VV task, we found a strong correlation between accuracy and model–human similarity for all age bins. In addition, we examined whether we would see differences in the strength of this correlation in data from children of different ages. We found that worse-performing models tended to show response patterns that were more similar to those from younger children, whereas better-performing models showed response patterns more similar to those from older children and adults – captured by an interaction effect between accuracy and age in a linear mixed-effect model ($b$ = -0.057, $SE$ = 0.003, $p$ < .001). These results suggest that children's lexical representations are more similar to those instantiated in lower-performing multimodal models early in development, and gradually become more similar to higher-performing multimodal modals across middle childhood.

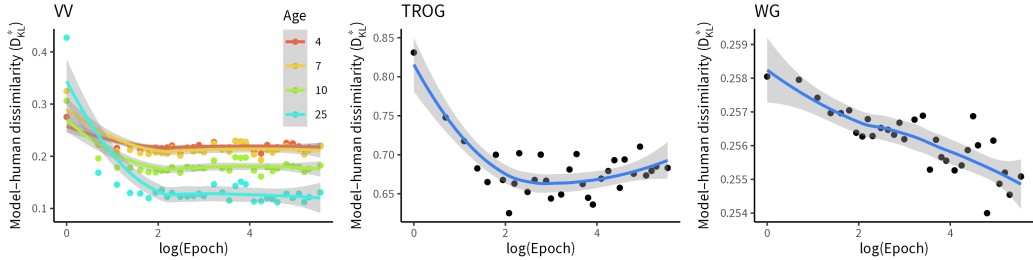

Figure 4: Trajectories of model–human similarity for VV, TROG, and WG. OpenCLIP-H becomes more human-like over training, and recovers developmental trends for VV.

## 5.2 Model training analysis

We next sought to understand whether human developmental trajectories were comparable to model training trajectories. We made use of OpenCLIP-H [53], an open model for which checkpoints were available on Hugging Face, allowing us to conduct "developmental" analyses. To do so, we sampled 32 checkpoints at approximately logarithmic intervals, and calculated model–human similarities for each task for each checkpoint. Training trajectory curves for three tasks (VV, TROG, and WG) are shown in Figure 4.

Overall, OpenCLIP increased in similarity to humans across training for these three tasks. Further, model performance on the Visual Vocabulary task also reflects a developmental trend: OpenCLIP exhibited greatest similarity to younger humans earlier in the training regime, and greatest similarity to older humans later in the training regime. This pattern matches the trend shown across models in Figure 3b – in both cases, better performing models show better correspondence to human data.

For brevity, we briefly describe the trajectory curves for OpenCLIP-H for the four remaining tasks; complete results are detailed in the SI. For LWL, we found that model–human similarity also improves with training, as expected. For the semantics tasks, the results were more complex, however. For WAT, model–human similarity largely remains constant over training, indicating that language representations do not significantly change. For VOC, the developmental trajectories were non-monotonic. For THINGS, model–human similarity *decreases* with training, perhaps indicating divergence with human visual representational space. Together, these results suggest that multimodal model representations may be less applicable to the semantics tasks we selected, and future work is needed to understand differences between semantic representations in uni- and multimodal representations.

## 5.3 Item-level analysis

Finally, we examined specific items to determine which items exhibited the greatest and least dissimilarity in response pattern between models and humans. For each model, we $z$-scored the $D^*_{\text{KL}}$ values to control for inter-model variation in overall model–human similarity. We then averaged $D^*_{\text{KL}}$ values across models for each trial, and extracted the top five items with the greatest mean divergence and the top five items with the least mean divergence for qualitative analysis. We illustrate this method for VV and WG, as shown in Table 3.

The most dissimilar items reveal certain features which may particularly drive model–human dissimilarity, particularly in comparison to the most similar items. For VV, some such features include polysemous targets (e.g., "horn" and "net"), targets which may have other labels (e.g., "hand plow" for "hoe", "pudding" for "flan"), or targets which could be labelled as one of the distractor categories (e.g., "lollipop" is a type of "candy"). For WG, several of the most dissimilar items have genuinely contentious captions – notably, the image for the caption "the dog is swimming and the person is standing" has the person hunched over rather than fully upright, the image for the caption "a person sits and a dog stands" has the dog mid-leap, and the image for the caption "green pants and blue top" has a top that could be described as grey. More broadly, across both VV and WG, humans appear to be better able to handle ambiguity and choose the most likely answer (perhaps through pragmatic reasoning [55]), whereas models are more likely to put less density on the true target.

Table 3: Top five most dissimilar items and top five most similar items between humans and all models for Visual Vocabulary and Winoground tasks.

| | |
|---|---|
| *Most dissimilar items* | |
| VV | horn (distractors: bone, chin, ladybug) |
| | hoe (distractors: peg, dustpan, beaker) |
| | flan (distractors: fuse, amplifier, turnstile) |
| | net (distractors: tee, domino, hydrant) |
| | lollipop (distractors: candy, doorbell, crumb) |
| WG | a person whispering into a dog's ear / a dog whispering into a person's ear |
| | there are more ladybugs than flowers / there are more flowers than ladybugs |
| | the dog is swimming and the person is standing / the dog is standing and the person is swimming |
| | blue pants and green top / green pants and blue top |
| | a person sits and a dog stands / a person stands and a dog sits |
| *Most similar items* | |
| VV | foam (distractors: float, quilt, asparagus) |
| | saddle (distractors: handle, figurine, broccoli) |
| | stump (distractors: log, bookshelf, showerhead) |
| | sorbet (distractors: palette, tamale, chive) |
| | typewriter (distractors: printer, sunglasses, drumstick) |
| WG | a horse getting wet / getting a horse wet |
| | a large living thing in front of a large non-living thing / a large non-living thing in front of a large living thing |
| | a deer's nose is resting on a child's hand / a child's hand is resting on a deer's nose |
| | clothing on lines / lines on clothing |
| | soft shoes are on a smooth floor / smooth shoes are on a soft floor |

## 6 Discussion

In this work, we introduced DEVBENCH, a multimodal benchmark for language learning consisting of seven tasks with corresponding data from both children and adults. Evaluating a set of vision–language models revealed variation across models in terms of model–human similarity across lexical, syntactic, and semantic domains. Furthermore, model–human similarity was strongly correlated with model accuracy, as well as model size to a lesser extent. Analysing OpenCLIP-H checkpoints across training also recovered developmental trends for some tasks, but not others.

More broadly, DEVBENCH provides a method for models trained on developmentally realistic data to be evaluated using a method that is comparable to how children are evaluated. Recent work has seen a plethora of new unimodal or multimodal learning models, including some that are evaluated here (e.g., [4]), but to date no models trained on developmentally realistic data (e.g., head-mounted camera data) have been evaluated actual developmental performance. We believe that benchmarks like DEVBENCH are essential for understanding the degree to which any given model can be used to approximate human learning.

Systematically comparing models and children's response patterns – rather than overall accuracy – is an essential piece of this puzzle. In doing so, our results already highlight ways in which model training trajectories are rather *unlike* human development trajectories, pointing towards new avenues for future work. For example, models appear to be worse than humans at handling ambiguous inputs; ambiguity resolution is thus a potential avenue for future multimodal model development.

### 6.1 Limitations and future work

The development of DEVBENCH was constrained by currently available human data; notably, this means that some tasks have relatively few items (in comparison with many other machine learning benchmarks), and some tasks have relatively few human participants. These limitations may result in uncertain reliabilities for the obtained results. More data are currently being collected for some of the tasks in DEVBENCH, which we anticipate will improve reliability, and simultaneously permit more expansive developmental analyses due to an extension of the included age ranges. However, we hope that this approach and standardized format will enable future researchers to contribute novel datasets to DEVBENCH, which we anticipate may grow in the coming years. Indeed, DEVBENCH

also includes primarily tasks for English-speaking children and adults, due to dataset availability. This situation reflects inequalities in language acquisition research [56], and more data is needed to construct a multilingual version of DEVBENCH, which will be more comprehensive and generalisable.

Additionally, model performance in our evaluation setup may be affected by the domain gap between models' training data and the stimuli used in our benchmark; for example, TROG uses cartoon depictions of events, which are dissimilar to the more photorealistic training data of CLIP. Thus, our evaluation results likely represent a lower bound on model–human similarity. Children as young as two years of age are able to learn from and generalise to pictographic depictions of objects [57–59], however, suggesting that generalisation across representations is an early-acquired skill.

In our analyses of the relationship between model–human similarity and model features, we could not hold model architectures constant due to the limited availability of relevant model checkpoints – thus we cannot make strong claims about precisely what aspects of models lead to better fit to human responses. Systematic evaluation of the roles of training data and model size thus remains an important research question for future work (see [60, 61] for related work on scaling).

Finally, we chose one linking hypothesis between model logits and human distributions, namely optimised KL divergence. This hypothesis assumes a perfect calibration between model logits and true uncertainties, which is only valid to some extent. Further research in model calibration [62] may help to mitigate these effects.

## 6.2 Conclusion

We hope that DEVBENCH can serve as an encouragement for machine learning researchers and cognitive scientists to develop vision–language models that not only perform well, but also can more closely approximate human learning. In particular, DevBench highlights the need for more fully open models with training checkpoints [63], enabling the study of training trajectories, as well as the need for more human-realistic training [2, 60] to better characterise model–human correspondences across developmental change. It may also be possible to adopt a developmental, multimodal approach to studying domains other than language, such as mathematical, logical, and social reasoning; more intentional data collection across a wide range of ages, tasks, and contexts will help to provide an increasingly comprehensive set of comparison data to better understand model learning trajectories (e.g., [64]). These research directions, among others, will help us to better understand the processes underlying human development, and how we might transfer humans' learning efficiencies onto machine learning models.

## Acknowledgments and Disclosure of Funding

Thanks to the Jacobs Foundation for support of stimulus and dataset creation. This work was funded in part by an NIH K99HD108386 award to BL and a Stanford Interdisciplinary Graduate Fellowship to WAM.

The authors made the following contributions. AWMT: Conceptualisation, Data curation, Methodology, Formal analysis, Software, Writing – original draft, Writing – review & editing. SY: Data curation, Software, Writing – original draft, Writing – review & editing. BL, WAM, TM, RDS, JDY: Data curation, Writing – review & editing. MCF: Conceptualisation, Methodology, Writing – original draft, Writing – review & editing, Supervision.

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

Table 4: Participant demographics for all tasks. A: Adult.

| Task | Age bin | Mean age | N | Country of origin | Test language |
|---|---|---|---|---|---|
| LWL | 1.5 | 1.42 | 166 | USA | English |
|  | 2 | 2.49 | 28 | USA | English |
|  | 2.5 | 2.56 | 100 | Australia | English |
| VV | 4 | 5.5 | 178 | USA | English |
|  | 7 | 8.22 | 458 | USA | English |
|  | 10 | 10.17 | 939 | USA | English |
|  | A |  | 205 | USA | English |
| TROG | 11 | 11.77 | 514 | USA | English |
| WG | A |  | 171 | Not specified | English |
| WAT | 5 | 5.6 | 200 | USA | English |
|  | 6 | 6.55 | 280 | USA | English |
|  | 8 | 8.73 | 280 | USA | English |
|  | 10 | 10.45 | 280 | USA | English |
|  | A |  | 6000 | USA | English |
| VOC | 0.3 | 0.33 | 24 | France | French |
|  | 0.8 | 0.9 | 24 | France | French |
|  | 1.6 | 1.6 | 25 | France | French |
| THINGS | A |  | 12340 | Not specified | English |

# A  Additional method details

## A.1  Additional task details

A breakdown of the demographics for each age bin for each task is shown in Table 4. An overview of key methodological dimensions for the sources of each task is shown in Table 5.

## A.2  Evaluated models

**CLIP-base & CLIP-large**  Contrastive Language–Image Pretraining [48] was one of the earliest families of vision–language models. It uses ViT for its vision encoder, and a GPT-style transformer for its text encoder, with further contrastive training on image–text pairs to increase the cosine similarity of matched image–caption pairs. We used the ViT-B/32 model for CLIP-base, and the ViT-L/14 model for CLIP-large.

**ViLT**  Vision-and-Language Transformer [49] introduced a method for vision–language pretraining without convolutions. It uses word embeddings concatenated with a linear projection of image patches as inputs, which are passed through a unified transformer encoder trained on three objectives: image–text matching, masked language modelling, and word–patch alignment. We used the ViLT-B/32 model in our evaluation.

**FLAVA**  Foundational Language and Vision Alignment Model [50] combines multimodal and unimodal pretraining objectives to broaden the types of usable data sources. The FLAVA model uses a ViT vision encoder, a ViT text encoder, and a ViT multimodal encoder that fused the vision and text hidden states. The vision encoder was trained on masked image modelling while the text encoder was trained on masked language modelling. The multimodal encoder was trained on masked multimodal modelling and image–text matching.

**BLIP**  Bootstrapping Language–Image Pretraining [51] uses a multimodal mixture of encoder–decoder transformers. It incorporates unimodal language and text encoders trained on image–text contrastive learning, an image-grounded text encoder trained on image–text matching, and an image-grounded text decoder trained on language modelling. They also used a bootstrapping approach to produce synthetic captions for web images, and noisy image–caption pairs are filtered out. We used the BLIP-B/16 model with image–text matching training on the COCO dataset [65] for evaluation.

**BridgeTower**  BridgeTower [52] adds more extensive cross-modal interaction to the "Two Tower" approach of image and text modelling. It includes a ViT vision encoder, a RoBERTa text encoder, and a cross-modal encoder connected to the unimodal encoders via bridge layers. The model is pretrained on masked language modelling and image–text matching.

**OpenCLIP-H**  OpenCLIP [53] is an open-data, open-source implementation of the CLIP model. We used the OpenCLIP-H/14 model trained on LAION-2B [66] for evaluation, which additionally has most intermediate checkpoints uploaded to Hugging Face; these were used for our training trajectory analyses.

**SigLIP**  SigLIP [54] is a modification of CLIP which does not rely on batch-wise normalisation, using a sigmoid loss instead of a softmax loss. It uses a ViT vision encoder and a transformer text encoder, and treats each image–text pair as a binary classification problem (match vs non-match).

**CVCL**  Child's View for Contrastive Learning [4] is a family of models trained on naturalistic egocentric videos drawn from head-mounted camera footage from a single child aged 6–25 months. We used the main model of CVCL, which uses a ResNeXt-50 model as its vision encoder, and mean-pooled text embedding as its text encoder; these were trained with a contrastive learning objective on co-occurring utterance–frame pairs from the egocentric video dataset.

## A.3  Evaluation setup

For lexicon and syntax tasks, models were evaluated by passing in each image–text pair for each trial as inputs, and obtaining model logits for each pair. For LWL and WG, there were two images in each trial, while for VV and TROG, there were four images in each trial. Model logits were then used to calculate the softmax-optimised KL divergence with human responses.

For VOC and THINGS, we obtained image embeddings for each stimulus, and obtained a representational similarity matrix (RSM) by calculating the pairwise cosine similarity for each pair of images. We then compared the model RSM with that obtained from human responses by calculating the Spearman's rank correlation coefficient for entries below the main diagonal in model and human RSMs.

For WAT, we obtained text embeddings for each stimulus, and calculated the pairwise cosine similarity for all cue–target pairs in the human response data. Model similarity values were then used to calculate the softmax-optimised KL divergence with human response distributions for each cue word.

Some models (e.g., BridgeTower) always required both image and text inputs. For these models, we used an empty string as the dummy text input when obtaining image embeddings for VOC and THINGS, and we used a neutral gray square as the dummy image input when obtaining text embeddings for WAT.

## A.4  Softmax-optimised Kullback–Leibler divergence

We used the softmax-optimised KL divergence as our novel metric of model–human dissimilarity. (Ordinary) KL divergence reflects how different a target probability distribution is from a reference probability distribution, often considered the true distribution. In the case of DEVBENCH, the reference distribution is obtained from human responses, while the target distribution is obtained from model responses.

The typical method of deriving probabilities from model responses is by conducting a softmax over logits. However, we considered that model logits may not be calibrated to the same scale as human responses, and therefore included the temperature, $\beta$, as a free parameter. In other words, the resultant distribution after optimisation can be considered a one-parameter projection from logit space to probability space, and the best fitting projection is that which induces the minimum KL divergence to the human response distribution.

# B  Additional detailed results

## B.1  Detailed benchmark results

Benchmark results for all tasks, split by human age bins, are shown in Table 6. As with the summarised results, CLIP-large and OpenCLIP-H perform relatively well across tasks, along with BridgeTower. CVCL performs poorly on most tasks, but exhibits the best correlations to 10-month-olds and 19-month-olds on the visual object categorisation task.

Additionally, although it is not the key index for our benchmark, we also report the accuracies on all datasets in Table 7 for reference.

## B.2  Detailed feature analyses

Correlations between model features and model–human similarities for all tasks, split by human age bins, are shown in Table 8. Note that semantics tasks have no "correct" answer, and thus no accuracies. Accuracy is consistently the most correlated feature even when results are broken down by age bins. The semantic tasks also show greater variability when considering age bins, such that some age bins have better correlations with size, and other age bins have better correlations with training. However, it is important to note that there was very little variability in model–human similarity for the word association task, which may have exaggerated the variability in feature–similarity correlations.

## B.3  Detailed trajectory analyses

OpenCLIP-H training trajectories for LWL, WAT, VOC, and THINGS are shown in Figure 5. LWL shows the expected developmental trend, with human-likeness increasing over training, although it is important to note that the three age groups use different sets of stimuli and are thus not inter-comparable. WAT shows almost no change over training, suggesting that language representations are not significantly changing over contrastive training. VOC shows a more complex pattern, whereby OpenCLIP-H similarity to infants aged 10 and 19 months exhibits a U-shaped pattern with a minimum around epoch 16, whereas similarity to infants aged 4 months exhibits an inverse U-shaped pattern also peaking around epoch 16. Finally, THINGS shows an inverse pattern, whereby model–human similarity actually decreases over training, rather than increasing. The trajectories for VOC and THINGS suggest that multimodal contrastive learning may actually result in representations that are less human-like, suggesting that human visual representations may not be shaped by vision–language correspondences to the same extent as models like OpenCLIP-H.

# C  Evaluating generation models

## C.1  Evaluation methods

As an exploratory analysis, we also evaluated generation models using two methods. The first method relied on next-token prediction. We passed in an input consisting of the image and a text prompt: "Caption: `<text>`'. Does the caption match the image? Answer Yes or No." This prompt closely matched the actual human task for the Winoground dataset. We obtained next-token prediction logits for 'Yes' and 'No', and then subtracted the 'No' logits from the 'Yes' logits, which approximates the log odds ratio between 'Yes' and 'No' for each image; these logit differences were then treated as image–text matching scores.

The second method relied on log-likelihood measurement. We passed in an input consisting of the image and a text prompt: "Describe this image. `<sep>` `<text>`" We then measured the log-likelihood of the sequence; these logits were then treated as image–text matching scores. Note that even though we measured the log-likelihood of the whole sequence, the substring that did not directly correspond to the target text was constant across comparisons for each trial, and thus should not affect the relative likelihood values.

We further attempted a third evaluation method relying on explicit ratings. We passed in an input consisting of the image and a text prompt: "Caption: `<text>`'. How much does the caption match the image? Give only a rating from 0 to 100." However, this method largely produced '0' or '100' responses, and thus we do not include the corresponding results.

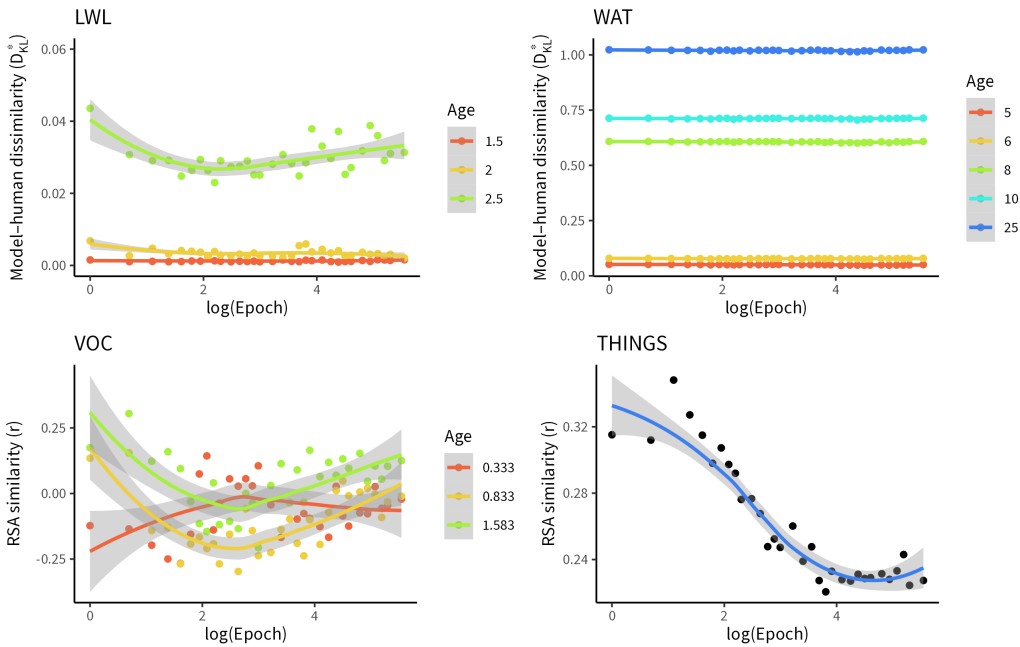

Figure 5: Trajectories of model–human similarity for LWL, WAT, VOC, and THINGS. Note that the three age groups for LWL are not comparable because they use different stimuli sets. For LWL and WAT, smaller values indicate greater model–human similarity, whereas for VOC and THINGS, larger values indicate greater model–human similarity.

Additionally, we only evaluated the lexical and syntactic tasks for these models, since it was not always clear that pure image or text features (needed for the semantic tasks) were extractable from the models.

## C.2   Evaluated models

**LLaVA-NeXT**   LLaVA-NeXT [67] is the 1.6 version of the Large Language and Vision Assistant (LLaVA) [68]. The model was trained on visual instruction data generated by GPT, using a visual encoder with Vicuna as the LLM component. LLaVA-NeXT improves upon LLaVA by increasing the image resolution and improving the mixture of the training data. We used the LLaVA-NeXT model with a Mistral 7B LLM.

**TinyLLaVA**   TinyLLaVA [69] introduced a framework for generating small vision–language models, using the core idea from LLaVA of having an image encoder along with an LLM. We used their best performing model, which is the 3.1B model using Phi-2 as the LLM and SigLIP as the training objective.

**Kosmos-2**   Kosmos-2 [70] is a multimodal large language model that incorporates object semantics (using bounding boxes) as well as grounding for text. It includes an image encoder as well as a Magneto transformer as its LLM component. The model was trained on image–text pairs, and was also instruction-tuned on image–text instruct data along with grounded image–text data.

**moondream2**   moondream2 [71] is a tiny vision–language model designed to be performant even with a small number of parameters, such that it can be run even on mobile devices. It includes an image encoder as well as Phi-2 as the LLM, and is trained on visual question answering. There is limited documentation about this model and its training method.

**CogVLM**   CogVLM [72] is a vision–language foundation model that allows for deep fusion of image and text representations by including a visual expert in the attention and feedforward layers of

the language model. It uses a ViT for its vision encoder, and a pretrained GPT as its LLM component. We used the CogVLM-17B model in our evaluation.

## C.3 Results and discussion

Results from the next-token prediction and log-likelihood measurement methods are shown in Table 9. In both methods, TinyLLaVA outperforms other generation models in most tasks, often by a sizeable margin. We also report accuracies from both methods in Table 10. We note that both in terms of model–human similarity and accuracy, the results may be underestimations as the specific prompts may be out of distribution for these models.

As we only evaluated five generation models, it was not possible to do further analyses relating to model features; further work evaluating a larger range of models would enable such analyses. It is also important to note that our evaluation methods represent only two out of a range of possible approaches; a more systematic search might reveal other aspects of variation across models.

Table 5: Overview of selected methodological dimensions for all tasks. A: Adult.

| Task | Source | Age bins | Recruitment method | Administration method | Data collection method | Experimental setup | Primary goal of study |
|---|---|---|---|---|---|---|---|
| LWL | Adams et al. | 1.5 | Hospital and others | In-person | Eyetracking | Preferential looking | Investigate associations between caregiver talk and language skills in full/preterm children |
|  | Frank et al. | 2 | Museum | In-person | Eyetracking | Preferential looking | Investigate the reliability of using tablets to collect experimental data from young children |
|  | Donnelly & Kidd | 2.5 | Not specified | In-person | Eyetracking | Preferential looking | Investigate the relationship between phonological onset density and lexical access |
| VV | Long et al. | 4, 7, 11, A | Schools and online | In-person and online | 4AFC | Best match | Investigate developmental changes in visual concept knowledge |
| TROG | Silverman & Yeatman | 11 | Schools | In-person | 4AFC | Best match | Investigate developmental changes in grammatical knowledge |
| WG | Thrush et al. | A | Amazon MTurk | Online | 2AFC | Match or non-match | Investigate visuo–linguistic compositionality in vision–language models |
| WAT | Entwisle | 5, 6, 8, 10 | Schools | In-person | Oral response | Word association | Investigate chidlren's verbal conceptual space and its relationship with demographic factors |
|  | Nelson et al. | A | Not specified | In-person | Written response | Word association | Investigate adults' verbal conceptual space and cue-to-target association strengths |
| VOC | Spriet et al. | 0.3, 0.8, 1.6 | Not specified | In-person | Eyetracking | Preferential looking | Investigate infants' visual conceptual space and its relationship with adult brain activity |
| THINGS | Hebart et al. | A | Amazon MTurk | Online | 3AFC | Odd one out | Investigate adults' visual conceptual space and its relationship with adult brain activity |

Table 6: Model performance across all tasks, split by human age bins (in years). A: Adult.

| Model | Lexicon | | | | | | | Syntax | |
|---|---|---|---|---|---|---|---|---|---|
| | LWL (↓) | | | VV (↓) | | | | TROG (↓) | WG (↓) |
| | 1.5 | 2 | 2.5 | 4 | 7 | 10 | A | 11 | A |
| CLIP-base | 0.002 | 0.007 | 0.032 | 0.220 | 0.228 | 0.195 | 0.177 | 0.732 | 0.256 |
| CLIP-large | 0.002 | 0.007 | 0.031 | 0.216 | **0.214** | **0.174** | **0.113** | 0.692 | 0.256 |
| ViLT | 0.003 | 0.005 | 0.018 | 0.248 | 0.304 | 0.293 | 0.460 | 0.682 | 0.252 |
| FLAVA | 0.001 | 0.006 | 0.031 | 0.214 | 0.220 | 0.190 | 0.166 | 0.912 | 0.254 |
| BLIP | **0.001** | 0.008 | 0.021 | 0.226 | 0.216 | 0.182 | 0.147 | **0.576** | 0.259 |
| BridgeTower | 0.001 | 0.005 | **0.017** | **0.213** | 0.245 | 0.231 | 0.369 | 0.584 | **0.250** |
| OpenCLIP-H | 0.002 | **0.002** | 0.031 | 0.220 | 0.219 | 0.183 | 0.131 | 0.683 | 0.255 |
| SigLIP | 0.051 | 0.020 | 0.131 | 0.426 | 0.578 | 0.587 | 0.857 | 0.888 | 0.258 |
| CVCL | 0.005 | 0.027 | 0.147 | 0.468 | 0.655 | 0.667 | 1.170 | 0.911 | 0.258 |

| Model | Semantics | | | | | | | | THINGS (↑) |
|---|---|---|---|---|---|---|---|---|---|
| | WAT (↓) | | | | | VOC (↑) | | | |
| | 5 | 6 | 8 | 10 | A | 0.3 | 0.8 | 1.6 | A |
| CLIP-base | 0.052 | 0.079 | 0.608 | 0.714 | 1.023 | -0.372 | -0.077 | 0.207 | **0.397** |
| CLIP-large | 0.052 | 0.079 | 0.608 | 0.714 | 1.023 | -0.250 | -0.038 | 0.302 | 0.246 |
| ViLT | 0.052 | 0.079 | 0.608 | 0.714 | 1.023 | -0.106 | -0.020 | -0.033 | 0.127 |
| FLAVA | 0.052 | 0.079 | 0.608 | 0.714 | 1.023 | -0.426 | 0.094 | 0.207 | 0.189 |
| BLIP | 0.051 | **0.079** | 0.608 | 0.714 | 1.023 | -0.078 | -0.237 | 0.002 | 0.185 |
| BridgeTower | 0.052 | 0.079 | **0.608** | **0.713** | 1.023 | -0.330 | -0.064 | 0.108 | 0.345 |
| OpenCLIP-H | **0.051** | 0.079 | 0.608 | 0.714 | **1.023** | **-0.021** | -0.010 | 0.125 | 0.227 |
| SigLIP | 0.052 | 0.079 | 0.608 | 0.714 | 1.023 | -0.179 | -0.068 | 0.161 | 0.192 |
| CVCL | 0.052 | 0.079 | 0.608 | 0.714 | 1.023 | -0.198 | **0.189** | **0.423** | 0.175 |

Table 7: Model accuracies across all tasks, averaged across ages.

| Model | Lexicon | | Syntax | |
|---|---|---|---|---|
| | LWL | VV | TROG | WG |
| CLIP-base | 0.987 | 0.958 | 0.462 | 0.608 |
| CLIP-large | 0.987 | 0.966 | 0.372 | 0.594 |
| ViLT | 1.000 | 0.790 | 0.449 | 0.655 |
| FLAVA | 0.987 | 0.941 | 0.308 | 0.605 |
| BLIP | 1.000 | 0.958 | 0.603 | 0.497 |
| BridgeTower | 1.000 | 0.832 | 0.628 | 0.731 |
| OpenCLIP-H | 1.000 | 0.975 | 0.487 | 0.579 |
| SigLIP | 1.000 | 0.924 | 0.423 | 0.541 |
| CVCL | 0.592 | 0.328 | 0.231 | 0.281 |

Table 8: Correlations between model features and model–human similarities across all tasks. Task performance is split by human age bins (in years). A: Adult.

| Feature | Lexicon | | | | | | | Syntax | |
| | LWL | | | VV | | | | TROG | WG |
| | 1.5 | 2 | 2.5 | 4 | 7 | 10 | A | 11 | A |
| Accuracy | 0.907 | 0.971 | 0.993 | 0.965 | 0.987 | 0.993 | 0.998 | 0.928 | 0.822 |
| Size | 0.831 | 0.818 | 0.711 | 0.811 | 0.837 | 0.844 | 0.854 | 0.434 | 0.254 |
| Training | 0.486 | 0.598 | 0.410 | 0.553 | 0.602 | 0.627 | 0.694 | 0.136 | -0.065 |

| Model | Semantics | | | | | | | | THINGS |
| | WAT | | | | | VOC | | | |
| | 5 | 6 | 8 | 10 | A | 0.3 | 0.8 | 1.6 | A |
| Size | 0.631 | 0.334 | 0.564 | 0.391 | 0.737 | 0.045 | -0.426 | -0.320 | 0.250 |
| Training | 0.689 | -0.064 | 0.172 | -0.041 | 0.834 | 0.154 | -0.194 | -0.045 | 0.277 |

Table 9: Generation model performance across all tasks. Arrows indicate the direction of better performance (i.e., lower is better vs. higher is better). Bolded results indicate most human-like result on a task.

| Model | # params | # images | Lexicon | | | | | | | Syntax | |
| | | | LWL (↓) | | | VV (↓) | | | | TROG (↓) | WG (↓) |
| | | | 1.5 | 2 | 2.5 | 4 | 7 | 10 | A | 11 | A |
| *Next-token prediction method* | | | | | | | | | | | |
| LLaVA | 7B | 1.31M | 0.052 | 0.030 | 0.162 | 0.468 | 0.656 | 0.671 | 1.176 | 0.910 | 0.258 |
| TinyLLaVA | 3.1B | 102K | 0.035 | **0.002** | **0.064** | **0.246** | **0.257** | **0.229** | **0.188** | **0.410** | **0.202** |
| Kosmos-2 | 1.6B | 90M | 0.053 | 0.022 | 0.117 | 0.468 | 0.656 | 0.672 | 1.176 | 0.905 | 0.258 |
| moondream2 | 1.9B | NA | **0.023** | 0.007 | 0.155 | 0.450 | 0.634 | 0.643 | 1.126 | 0.757 | 0.254 |
| CogVLM | 17B | 1.5B | 0.059 | 0.029 | 0.149 | 0.447 | 0.612 | 0.619 | 1.063 | 0.868 | 0.237 |
| *Log-likelihood measurement method* | | | | | | | | | | | |
| LLaVA | 7B | 1.31M | 0.059 | 0.030 | 0.171 | 0.468 | 0.656 | 0.671 | 1.176 | 0.910 | 0.258 |
| TinyLLaVA | 3.1B | 102K | **0.043** | 0.017 | **0.075** | **0.402** | 0.560 | 0.562 | **0.971** | **0.738** | **0.220** |
| Kosmos-2 | 1.6B | 90M | 0.051 | **0.013** | 0.146 | 0.415 | **0.553** | **0.557** | 0.975 | 0.781 | 0.225 |
| moondream2 | 1.9B | NA | 0.059 | 0.030 | 0.174 | 0.468 | 0.656 | 0.671 | 1.175 | 0.910 | 0.258 |
| CogVLM | 17B | 1.5B | 0.059 | 0.018 | 0.174 | 0.468 | 0.657 | 0.672 | 1.177 | 0.908 | 0.234 |

Table 10: Model accuracies across all tasks, averaged across ages.

| Model | Lexicon | | Syntax | |
| | LWL | VV | TROG | WG |
| *Next-token prediction method* | | | | |
| LLaVA | 0.579 | 0.176 | 0.205 | 0.509 |
| TinyLLaVA | 1.000 | 0.958 | 0.782 | 0.728 |
| Kosmos-2 | 0.750 | 0.252 | 0.333 | 0.497 |
| moondream2 | 0.816 | 0.345 | 0.436 | 0.529 |
| CogVLM | 0.553 | 0.437 | 0.256 | 0.538 |
| *Log-likelihood measurement method* | | | | |
| LLaVA | 0.487 | 0.202 | 0.231 | 0.497 |
| TinyLLaVA | 0.855 | 0.462 | 0.474 | 0.614 |
| Kosmos-2 | 0.632 | 0.479 | 0.436 | 0.547 |
| moondream2 | 0.461 | 0.210 | 0.154 | 0.474 |
| CogVLM | 0.461 | 0.235 | 0.179 | 0.500 |

