# OpenReview forum: "DevBench: A multimodal developmental benchmark for language learning"
_NeurIPS.cc/2024/Datasets_and_Benchmarks_Track — NeurIPS 2024 Track Datasets and Benchmarks Oral_

### Official Review · Reviewer_k3Ee · 2024-07-20
**Nice benchmark for comparing the developmental trajectories between humans and machines**

**Rating:** 7
**Confidence:** 5
**Correctness:** Yes, claims are correct and the bench…

**Review:**

Overall, I like this paper, and it's definitely a good step forward for making apples-to-apples comparisons between language models and humans. The benchmark itself is carefully constructed and is developmentally inspired, with concrete prior human studies as support. I will discuss strengths and weaknesses in further detail below.

**Strengths:**

- The paper is very well-written and has a good motivation. I enjoyed reading the introduction and the way the authors compare their work to previous works.

- The introduced BevBench actually highlights some alternative evaluation metrics beyond the commonly used accuracies. For example, it emphasizes the importance of comparing human-model response similarities and how the performance trajectories change over time. In other words, it focuses on how the models learn and behave rather than simply solve tasks. That's a great job.

- It tackles a wide range of tasks, focusing on lexical, syntactic, and semantic ability. Each task category is represented by one or multiple existing datasets/benchmarks, along with the ages a typical human can solve.

**Additional Feedback:**

Some minor formatting comments: according to neurips official guidelines, it is suggested not to use vertical rules in tables (e.g., Table 1 & 2).

**Clarity:**

Yes, the paper is well written. I like the overall motivation and the introduction section.

**Documentation:**

There might be a lack of sufficient maintenance details. For example, a dataset card may help. Meanwhile, the GitHub repo: ``github.com/alvinwmtan/dev-bench`` is not accessible by the reviewers during the review period.

The authors do mention why the repo is not accessible by now in lines 500-501: ``note that the repository is currently private as some of the unpublished datasets have not been released to the public.`` So, I wonder when the repo will be estimated to turn public?

**Ethics:**

No need for an ethics review.

**Limitations:**

Yes, the authors have adequately addressed the limitations of this paper. No foreseeable negative societal impact.

**Opportunities For Improvement:**

- One limitation is the choice of baseline models: some of the most advanced vision-language models are not included (e.g., lava-1.5, GPT-4o, or instructblip; for GPT-4o, it might be hard to get logits, but sampling at a higher temperature and then count the frequency might be a solution). Meanwhile, the model architectures are not fixed when comparing the performance - amount of training data / model size correlations. A better way might be using openCLIP intermediate checkpoints.

- It's hard to interpret these results, especially the KL divergence-based measurements. For example, almost all the models score 0.494-0.495 on the WAT task (the WG task also behaves similarly); what does that mean? Are the models collapsing? I am also curious how well a random baseline would perform on these tasks.

**Relation To Prior Work:**

Yes, related works are clearly discussed.

**Summary And Contributions:**

This paper introduces DevBench, which comprises seven tasks covering lexical, syntactic, and semantic evaluations for language models. These tasks are carefully chosen based on the developmental trajectories of human language learning and try to measure machine-human similarities for building better language models. Evaluations show that machine-human similarities correlate with several other features and may inspire how machine and human language learning match and diverge.

---

> ### Author Rebuttal · Authors · 2024-08-16
>
> Thank you for your thoughtful remarks and suggestions, and we appreciate that you mentioned how our evaluation allows for better comparisons between models and humans. We respond to the review in detail below.
>
> **Q1: One limitation is the choice of baseline models: some of the most advanced vision-language models are not included (e.g., lava-1.5, GPT-4o, or instructblip; for GPT-4o, it might be hard to get logits, but sampling at a higher temperature and then count the frequency might be a solution). Meanwhile, the model architectures are not fixed when comparing the performance - amount of training data / model size correlations. A better way might be using openCLIP intermediate checkpoints.**
>
> A1: We have now included some newer models including LLaVA; please see the general comment for more details.
>
> We appreciate that model architectures were not fixed when conducting the model feature analysis; however, it is difficult to do so without doing additional training (as number of training steps may not match number of images seen, since images may be repeated across epochs). We agree that systematically evaluating the role of training data and model size is important, and have included it as a possible future direction:
>
> > It is important to note that our analyses of the relationship between model–human similarity and model features did not hold model architectures constant due to the limited availability of relevant model checkpoints which would have enabled such analyses. Systematic evaluation of the roles of training data and model size nonetheless remains an important research question for future work.
>
> **Q2: It's hard to interpret these results, especially the KL divergence-based measurements. For example, almost all the models score 0.494-0.495 on the WAT task (the WG task also behaves similarly); what does that mean? Are the models collapsing? I am also curious how well a random baseline would perform on these tasks.**
>
> A2: We added a human baseline for tasks on which we had participant-level data (LWL, VV, TROG, VOC), and also included a random baseline as a negative baseline; please see the general comment for more details. The WAT results are indeed more difficult to interpret; they may indicate that text embedding spaces are very similar across models—notably, WAT performance also remains relatively constant across OpenCLIP training, as shown in Figure 5.
>
> **Q3: There might be a lack of sufficient maintenance details. For example, a dataset card may help. Meanwhile, the GitHub repo: github.com/alvinwmtan/dev-bench is not accessible by the reviewers during the review period; I wonder when the repo will be estimated to turn public?**
>
> A3: Apologies for this, we were working on licensing for the datasets but this is now resolved. We have made the repo accessible and all the data are contained there—the link should be live now. We will also add a dataset card to the repository.
>
> **Q4: According to neurips official guidelines, it is suggested not to use vertical rules in tables (e.g., Table 1 & 2).**
>
> A4: Thank you for this note—we have updated our table format.

---

> > ### Comment · Reviewer_k3Ee · 2024-08-30
> >
> > Thanks for the response. My score remains the same and I recommend acceptance.

---

### Official Review · Reviewer_fso1 · 2024-07-24
**Good cogsci benchmark with interesting results**

**Rating:** 7
**Confidence:** 3
**Correctness:** There are no concerns on the technica…
**Clarity:** Yes. The paper is well written.

**Review:**

I don’t have serious concerns with the paper as I see more of its benefits with the release of the unpublished datasets will outweigh its risks, if any. However, some aspects of the paper can be improved:

The specific details from the experiment setup should be included in the appendix. This is to make the entire paper a well-defined resource about the compiled datasets. Include information about the demographics of the users (if there are), more information about the setup of each experiment, main goal of its source paper, etc. Section 3.1 is just a summary of the main information about the datasets but this should be expanded in the Appendix.

Is this softmax optimized KL divergence method originally used here in the paper or have other papers used this previously? I would like to read more if this is a novel idea or adopted from some work. Moreover, it would be good for the paper to also mention what method they used for calculating the (dis)similarity between human and model responses and how the KL divergence differs from their approach.

The conclusion section is weak and can be improved. Discuss how the existence of the benchmark (as well as the release of some of the unpublished dataset component) paves way for new research directions in both ML and cognitive science as well as how it can be improved and developed by the open research community.

**Strengths:**

The paper is very easy to read and follow which is important for wider adoption of the study’s contributions.

The paper has a well-discussed motivation and the need for a good benchmark of multimodal learning that reflects how children learn is indeed important to advance the field of understanding how ML models work. This paper address a real problem and the existence of the benchmark and dataset will open new opportunities in ML x cognitive science research.

The contribution dataset is compiled from existing and published datasets on various linguistic types, thus, can be helpful as they can be considered new.

The results from the benchmarking process of multimodal models using the data are interesting. It’s interesting to know that the model-human similarity is correlated with accuracy but not data size, and that there is a connection with children’s lexical representations with lower-performing multimodal models and for high-performing models with adults.

**Additional Feedback:**

See main review.

**Documentation:**

The benchmark is open via Github.

**Ethics:**

No concerns.

**Limitations:**

The limitations seem aptly discussed.

**Opportunities For Improvement:**

See main review.

**Relation To Prior Work:**

No concerns.

**Summary And Contributions:**

The paper describes DEVBENCH, a multimodal benchmark for further understanding how language models correlate with human learning through a series of language evaluation tasks spanning the domains of lexical, syntactic, and semantic ability. The benchmark includes a composition of behavioral data from various published and unpublished works. The authors motivate the need for this benchmark (and the dataset) by explaining that there is a “data gap” in the form of learning differences between human language learners and models as the latter typically requires magnitude of data to learn than the former. The authors also state that “it is crucial to evaluate models on benchmarks that can indicate whether the language ability gained by machine learning models matches the language ability gained by children when exposed to similar developmental data.” Interesting results were obtained from the experiments with vision-language models with the language learning tasks including observing performance of models being more correlated with accuracy than size. Overall, I think this is a good benchmark and the release of its datasets will be useful for cognitive science and ML research.

---

> ### Author Rebuttal · Authors · 2024-08-16
>
> Thank you for your suggestions to improve the utility of our benchmark. We respond to the review in detail below:
>
> **Q1: The specific details from the experiment setup should be included in the appendix. This is to make the entire paper a well-defined resource about the compiled datasets. Include information about the demographics of the users (if there are), more information about the setup of each experiment, main goal of its source paper, etc. Section 3.1 is just a summary of the main information about the datasets but this should be expanded in the Appendix.**
>
> Thank you for this suggestion. We added tables for demographics (mean age, number of participants per age bin, country, test language) and for experimental characteristics (recruitment method, administration method, data collection method, experimental setup, primary goal of study) to the Appendix.
>
> **Q2: Is this softmax optimized KL divergence method originally used here in the paper or have other papers used this previously? I would like to read more if this is a novel idea or adopted from some work. Moreover, it would be good for the paper to also mention what method they used for calculating the (dis)similarity between human and model responses and how the KL divergence differs from their approach.**
>
> Softmax-optimised KL divergence is a novel metric, although it is very closely related to ordinary KL divergence. We have added a section in the Appendix to explain how our metric differs from KL divergence.
>
> > We used the softmax-optimised KL divergence as our novel metric of model–human dissimilarity. (Ordinary) KL divergence reflects how different a target probability distribution is from a reference probability distribution, often considered the true distribution. In the case of DevBench, the reference distribution is obtained from human responses, while the target distribution is obtained from model responses.
> >
> > The typical method of deriving probabilities from model responses is by conducting a softmax over logits. However, we considered that model logits may not be calibrated to the same scale as human responses, and therefore included the temperature, $\beta$, as a free parameter. In other words, the resultant distribution after optimisation can be considered a one-parameter projection from logit space to probability space, and the best fitting projection is that which induces the minimum KL divergence to the human response distribution.
>
> **Q3: The conclusion section is weak and can be improved. Discuss how the existence of the benchmark (as well as the release of some of the unpublished dataset component) paves way for new research directions in both ML and cognitive science as well as how it can be improved and developed by the open research community.**
>
> A3: Thanks for this invitation. Within our space constraints, we have added a bit more forward-looking text to this section, including moving some text downwards from the limitations section, as well as incorporating additional future directions:
>
> > **In particular, DevBench highlights the need for more fully open models with training checkpoints (Frank, 2024), enabling the study of training trajectories, as well as the need for more human-realistic training (Warstadt et al., 2023) to better characterise model–human correspondences across developmental change. These research directions, among others, will help us** to better understand the processes underlying human development, and how we might transfer humans’ learning efficiencies onto machine learning models.

---

> > ### Comment · Reviewer_fso1 · 2024-08-17
> > **Acknowledgment of response**
> >
> > Dear authors,
> >
> > Thank you for the response to my review. This is to acknowledge that I have read said response as well as the responses to my fellow reviewer's questions/feedback. My score is accept (7) as I think the dataset will be a good addition to ML x CogSci research. I trust the authors will accommodate my recommendations on improving the discussion of some parts of the paper as indicated in my feedback.

---

### Official Review · Reviewer_bnQw · 2024-07-25
**Evaliating models based on their alignment with humans at different stages of development (e.g., early adolescence vs. adulthood)**

**Rating:** 5
**Confidence:** 4

**Review:**

Please see Strengths and Opportunities for improvement.

In general, I hope the authors would put together a usable codebase and testbed for easy evaluation, increase the number of samples in the dataset, clarify the metrics, and add additional models.

**Strengths:**

S1. Out-of-the-box thinking applied to evaluation. I appreciate that the benchmark probes for abilities that develop, empirically, at different stages of adolescence and adulthood.

S2. Paper is clear in its scope.

S3. Some analysis both of converged models and models through training

**Additional Feedback:**

See above.

**Clarity:**

In some ways yes; however, key metrics for some tasks, which seem to be different than KL divergence are obscured.

**Correctness:**

The dataset is constructed transparently, but cobbling together existing datasets.

**Documentation:**

The main link: github.com/alvinwmtan/dev-bench does not seem to be live.

**Limitations:**

Yes, the paper is transparent with limitations.

**Opportunities For Improvement:**

W1. While the benchmark is potentially interesting for building an empirical understanding of capacity and capabilities, the benchmark does not seem particularly relevant for evaluating frontier models, which are expected to be at "adult" level intelligence.

W2. The link, github.com/alvinwmtan/dev-bench, does not seem to be live, making it hard to judge the reproducibility and usefulness of the benchmark to the larger community.

W3. How is a logit evaluation applicable to the "Lexicon: Looking-while-listening (LWL)" evaluation, which is based on time?

W4. If the metric is always KL divergence as suggested in 3.2, why is higher sometimes better for certain task? The metrics don't seem to be super clear to me, perhaps make this more clear.

W5. Given the low computational cost of the benchmark, I suggest testing even more models, e.g., LLaVA models, SigLip models, DataComp models, etc.

W6. What is  replacement stimuli?

W7. Table 2 numbers are not super interpretable to me. What is the KL divergence between pairs of humans? Maybe this can give a baseline to better interprate the results?

**Relation To Prior Work:**

There are many more benchmarks that can multimodal models can be used to tackle. Image classification comes to mind. Hence, I suggest citing many more prior works, perhaps in an extended related work section.

Additionally, Weihs et al. Benchmarking Progress to Infant-Level Physical Reasoning in AI. 2022.

For the "Model learning trajectory analyses", consider adding references related to how downstream performance changes throughout a training trajectory.

**Summary And Contributions:**

The paper proposes/contributes an evaluation across 3 core areas for developmental evaluation of multimodal models, probing for similarities with humans at various stages of developement.

---

> ### Author Rebuttal · Authors · 2024-08-16
>
> Thank you for your questions and suggestions for improving the clarity of our project. We respond to the review in detail below:
>
> **W1. While the benchmark is potentially interesting for building an empirical understanding of capacity and capabilities, the benchmark does not seem particularly relevant for evaluating frontier models, which are expected to be at "adult" level intelligence.**
>
> A1: Although it is true that the benchmark is primarily developmental in nature, we believe that there remain important reasons to benchmark frontier models:
>
> 1. Some of our tasks (VV, WG, THINGS, WAT) were assessed on adults. Thus, there is direct comparability between frontier models and adult behaviour, and it is still important to understand model–human similarity for adults to understand gaps in model performance (see for example Section 5.3).
>
> 2. Frontier models can also be evaluated in terms of their learning trajectories (much like our trajectory analysis in Section 5.2). Digging into model performance over the course of training gives us a better sense of how model and human learning processes may differ at a higher-order level, thereby suggesting ways in which model training (e.g., objectives, curricula) could be modified to increase similarity with human developmental trajectories.
>
> **W2. The link, github.com/alvinwmtan/dev-bench, does not seem to be live, making it hard to judge the reproducibility and usefulness of the benchmark to the larger community.**
>
> A2: Apologies for this, we were working on licensing for the datasets but this is now resolved. We have made the repo accessible and all the data are contained there—the link should be live now.
>
> **W3. How is a logit evaluation applicable to the "Lexicon: Looking-while-listening (LWL)" evaluation, which is based on time?**
>
> A3: LWL data can be analysed in a number of ways, including time profile, reaction time, and accuracy (Fernald et al., 2008). Here we used the last metric, which is operationalised as the proportion of time that an infant spends looking at the target (rather than the distractor) during the critical window. This metric is interpreted as the probability with which the infant correctly identifies the target, in the same way that softmaxed logits represent the probability that the model correctly identifies the target.
>
> **W4. If the metric is always KL divergence as suggested in 3.2, why is higher sometimes better for certain task? The metrics don't seem to be super clear to me, perhaps make this more clear.**
>
> A4: As noted in Section 3.2, as well as the first paragraph of Section 4, KL divergence was used for the lexicon and syntax tasks as well as WAT, but VOC and THINGS were assessed using RSA similarity, for which higher scores reflect greater similarity. We clarified the wording slightly in Section 3.2 to emphasise this:
>
> > For the visual semantic tasks, we **instead** conducted human–model comparison by applying representational similarity analysis (RSA) on human and model representational similarity matrices, **which represents correlations in the representational geometries of humans and models.**
>
> **W5. Given the low computational cost of the benchmark, I suggest testing even more models, e.g., LLaVA models, SigLip models, DataComp models, etc.**
>
> A5: We have added SigLIP and LLaVA, along with a few other models; please see the general comment for more details.
>
> **W6. What is replacement stimuli?**
>
> A6: These are new stimuli that we had to replace because the originals did not have a sharable license. We added some text to clarify this:
>
> > Some images in the original stimuli set were not sharable due to license restrictions; in these cases, we used replacement stimuli matched for visual and semantic properties.
>
> **W7. Table 2 numbers are not super interpretable to me. What is the KL divergence between pairs of humans? Maybe this can give a baseline to better interprate the results?**
>
> A7: We added a human baseline for tasks on which we had participant-level data (LWL, VV, TROG, VOC), and also included a random baseline as a negative baseline; please see the general comment for more details.
>
> **Q8: There are many more benchmarks that can multimodal models can be used to tackle. Image classification comes to mind. Hence, I suggest citing many more prior works, perhaps in an extended related work section. Additionally, Weihs et al. Benchmarking Progress to Infant-Level Physical Reasoning in AI. 2022. For the "Model learning trajectory analyses", consider adding references related to how downstream performance changes throughout a training trajectory.**
>
> A8: Thanks, we have added several related works, including the Weihs et al. reference given here. We give some background on multimodal benchmarking, but our primary aim is to provide a benchmark that is relevant for understanding similarities and differences in language understanding between models and human learners. Thus, we focused in our related work section on human comparison benchmarks, primarily in the domain of language.

---

> ### Comment · Reviewer_bnQw · 2024-08-30
> **Rebuttal review**
>
> Thanks for the clarification and the additional results. Upon review I am bumping my score to a 6. Note: edit functionality is currently disabled, but will do this once it is enabled.

---

### Official Review · Reviewer_6bM6 · 2024-07-25
**Interesting development language learning benchmark for evaluating multi-modal models**

**Rating:** 6
**Confidence:** 4

**Review:**

Pros:
- Clarity: The paper was well-written and well-motivated
- Significance:
   - The introduced benchmark is relevant to understanding and developing multi-modal models which is promising for future research
   - The dataset accommodates various tasks, well classified into different levels of difficulty
- Originality:
   - The benchmark and evaluation protocol is novel

Cons:
- Clarity: Unclear explanation of some important details (please see below in Improvement and Clarity sections)
- Originality:
   - The dataset is a combination of multiple established dataset

**Strengths:**

- The proposed benchmark is novel in the following aspects:
   - The dataset used as a testbed for the benchmark is collected from a wide range of human ages, from very young children to adults, with various levels of difficulty.
   - The dataset and the benchmark is divided into sub-tasks that cover these levels of difficulty in language understanding, which corresponds to children's language development at different ages.
   - The proposed metric compares performances of multi-modal models relative to human's performance instead of absolute performance.
- The analyses present in the paper yields interesting properties that would be of interest to the multi-modal research community.
- The proposed benchmark and dataset are well-motivated by developmental language learning.

**Additional Feedback:**

- Details about how the multi-modal models were evaluated should be discussed (as mentioned in Improvement section above)
- Please fix the GitHub link

**Clarity:**

The paper is well-written. All terminologies and concepts are explained thoroughly.

**Correctness:**

- The claims made in the submission are correct and backed up by evidence from experiments and analyses.
- The dataset was well-motivated and carefully designed for the benchmark
- There is a concern about domain gap between training/testing settings of the multi-modal models (details in Opportunities for Improvement), which can potentially make the performance of these methods unreliable.

**Documentation:**

GitHub link not accessible. Data collection was discussed in the paper. Licensing information is not clear.

**Ethics:**

No ethical concern

**Limitations:**

The authors discussed the limitations of the proposed work in detail

**Opportunities For Improvement:**

- The details for how the multi-modal models were evaluated are missing. For example: What are the inputs to OpenCLIP for TROG task and what are the outputs?
- Along the same line, a concern can be that these multi-modal models were trained on a different kind of data (e.g. CLIP was trained on pairs of image and image caption). This can lead to domain gap between the training data and the evaluation data (e.g. the visual encoder of these models might not be able to encode images with white background or with drawings well since they were trained on in-the-wild naturalistic images; or the language prompts they were trained on did not have the same format as the language prompts or responses expected for these tasks). Without careful mitigation of this domain gap, the performance of these models might degrade and become unreliable.
- Will the findings in the paper hold with the more advanced multi-modal models such as LLaVA (open-sourced), CogVLM (open-sourced), Kosmos-2 (open-sourced), GPT-4v, Gemini, etc.? Is there any specific reason why we should not evaluate DevBench tasks on these more advanced models?

**Relation To Prior Work:**

The paper discussed prior works carefully and the contributions of the work are significant with regard to prior contributions.

**Summary And Contributions:**

- The paper introduced a benchmark, DevBench to evaluate multi-modal models on language tasks that can be compared directly with human's performance.
- The dataset for this benchmark covers multiple levels of difficulty of language tasks, collected from both children of different ages and adult.
- The analyses on current multi-modal models present different aspects and behaviors that these models observe that are similar and diverge from human behaviors. These results open opportunities for future research to study and improve the training of these multi-modal models.

---

> ### Author Rebuttal · Authors · 2024-08-16
>
> Thank you for your thoughtful comments and suggestions, and we are glad that you appreciated the developmental motivation of our benchmark. We respond to the review in detail below:
>
> **Q1: The details for how the multi-modal models were evaluated are missing. For example: What are the inputs to OpenCLIP for TROG task and what are the outputs?**
>
> A1: Thanks for this comment. We have added more details on the evaluations in the Appendix:
>
> > For lexicon and syntax tasks, models were evaluated by passing in each image–text pair for each trial as inputs, and obtaining model logits for each pair. For LWL and WG, there were two images in each trial, while for VV and TROG, there were four images in each trial. Model logits were then used to calculate the softmax-optimised KL divergence with human responses.
> >
> > For VOC and THINGS, we obtained image embeddings for each stimulus, and obtained a representational similarity matrix (RSM) by calculating the pairwise cosine similarity for each pair of images. We then compared the model RSM with that obtained from human responses by calculating the Spearman’s rank correlation coefficient for entries below the main diagonal in model and human RSMs.
> >
> > For WAT, we obtained text embeddings for each stimulus, and calculated the pairwise cosine similarity for all cue–target pairs in the human response data. Model similarity values were then used to calculate the softmax-optimised KL divergence with human response distributions for each cue word.
> >
> > Some models (e.g., BridgeTower) always required both image and text inputs. For these models, we used an empty string as the dummy text input when obtaining image embeddings for VOC and THINGS, and we used a neutral gray square as the dummy image input when obtaining text embeddings for WAT.
>
> **Q2: Along the same line, a concern can be that these multi-modal models were trained on a different kind of data (e.g. CLIP was trained on pairs of image and image caption). This can lead to domain gap between the training data and the evaluation data (e.g. the visual encoder of these models might not be able to encode images with white background or with drawings well since they were trained on in-the-wild naturalistic images; or the language prompts they were trained on did not have the same format as the language prompts or responses expected for these tasks). Without careful mitigation of this domain gap, the performance of these models might degrade and become unreliable.**
>
> A2: This is an important comment, and it is true that model performance here likely represents a lower bound on model–human similarity. We have made an additional note in the limitations to acknowledge this:
>
> > Additionally, model performance in our evaluation setup may be affected by the domain gap between models’ training data and the stimuli used in our benchmark; for example, TROG uses cartoon depictions of events, which are dissimilar to the more photorealistic training data of CLIP. Thus, our evaluation results likely represent a lower bound on model–human similarity. It is nonetheless important to note that children as young as two years of age are able to learn from and generalise to pictographic depictions of objects (Ganea et al., 2008; Simcock & DeLoache, 2006; Tare et al., 2010), suggesting that generalisation across representations is an early-acquired skill.
>
> **Q3: Will the findings in the paper hold with the more advanced multi-modal models such as LLaVA (open-sourced), CogVLM (open-sourced), Kosmos-2 (open-sourced), GPT-4v, Gemini, etc.? Is there any specific reason why we should not evaluate DevBench tasks on these more advanced models?**
>
> A3: We have added some of these models (LLaVA, CogVLM, Kosmos-2); please see the general comment for more details.

---

> > ### Comment · Reviewer_6bM6 · 2024-08-31
> >
> > I appreciate the authors for their effort in addressing my comments. I will increase my score to 7.

---

### Author Rebuttal · Authors · 2024-08-16

We would like to thank the reviewers for their insightful comments, which have helped to improve this work. Here, we highlight three key changes that we have made to our paper.

**1. Repository availability**

We were previously working on licensing for some of our unpublished datasets, which have now been resolved; our GitHub repository is now live and can now be accessed at [github.com/alvinwmtan/dev-bench](https://www.github.com/alvinwmtan/dev-bench).

**2. New baselines**

We have added a human baseline for tasks on which participant-level data were available (LWL, VV, TROG, VOC). To estimate this baseline, we randomly split the participants into two groups and calculated the between-group softmax-optimised KL divergence or RSA similarity as appropriate, repeating for 1000 random splits. We used the median result as a point estimate of the human baseline. The human baseline serves as a positive baseline for our tasks.

We also added a random baseline for all tasks, generated using a random initialisation of the OpenCLIP model. The random baseline serves as a negative baseline for our tasks.

We anticipate that the two baselines will assist in interpretation of the results, demonstrating the dynamic range that is possible for each task.

**3. New models**

We also evaluated a set of newer models on our tasks to improve comprehensiveness. These models include SigLIP, as well as a set of conditional generation models (Kosmos-2, moondream2, TinyLLaVA, LLaVA, and CogVLM). The results for these models are presented in below.

Notably, because the conditional generation models do not assign a similarity score between images and texts, we used an alternative method of evaluation: We passed in a text prompt of the format “Caption: {text}. Does the caption match the image? Answer either Yes or No.”, and we extracted logits for “Yes” and “No”. This prompt closely matched the actual human task for the Winoground dataset. We then subtracted the “No” logits from the “Yes” logits, which approximates the log odds ratio between “Yes” and “No” for each image; these logit differences were then used to calculate the softmax-optimised KL divergence. Note that there is as yet limited consensus for the best method to obtain image–text matching scores for conditional generation models; we intend to attempt alternative methods (namely, using a captioning task and getting logits for the texts, or asking for explicit matching ratings from 0–100) to understand how task construction affects model–human similarity. We also only evaluated the lexical and syntactic tasks for these models, since it was not always clear that pure image or text features (needed for the semantic tasks) were extractable from the models.

| Model             | # params | # images | LWL (↓) | VV (↓) | TROG (↓) | WG (↓) | WAT (↓) | VOC (↑) | THINGS (↑) |
|-------------------|----------|----------|---------|--------|----------|--------|---------|---------|------------|
| SigLIP            | 800M     | 9B       | 0.067   | 0.612  | 0.888    | 0.258  | 0.495   | -0.028  | 0.192      |
| | | | | | | | | | |
| LLaVA             | 7B       | 1.31M    | 0.081   | 0.743  | 0.910    | 0.258  |         |         |            |
| TinyLLaVA         | 3.1B     | 102K     | 0.033   | 0.230  | 0.410    | 0.202  |         |         |            |
| Kosmos-2          | 1.6B     | 90M      | 0.064   | 0.743  | 0.905    | 0.258  |         |         |            |
| moondream2        | 1.9B     | NA       | 0.062   | 0.713  | 0.757    | 0.254  |         |         |            |
| CogVLM            | 17B      | 1.5B     | 0.079   | 0.685  | 0.868    | 0.237  |         |         |            |
| | | | | | | | | | |
| Human             |          |          | 0.010   | 0.091  | 0.028    |        |         | 0.251   |            |
| Random (OpenCLIP) | 1.0B     | 0        | 0.087   | 0.740  | 0.908    | 0.258  | 0.495   | 0.246   | 0.054      |

We hope that these changes, as well as additional changes detailed in our replies below, will help to clarify and improve DevBench, making it more comprehensible, comprehensive, and useful for AI researchers and cognitive scientists alike.

---

### Decision · Program_Chairs · 2024-09-26

**Decision:**

Accept (Oral)

**Comment:**

Summary:

This paper introduces DevBench, a benchmark designed to evaluate multimodal LMs on behavioral data from both children and adults. The benchmark consists of seven tasks designed to test different aspects of language understanding. It was used to evaluate a variety of models, showing that models' performance tends to be more similar to that of adults than that of children. The results were thoroughly analysed.

Contributions:

a. The DevBench benchmark

Summary of reviewers' opinions:

Two of the reviewers judged this paper a clear accept; for one reviewer it was marginally above threshold, and for another one it was marginally below.

Reviewers fs01 and k3Ee, who ranked the paper a clear accept, thought the paper well-written and thought the dataset was going to be very useful; they liked the choice of tasks and the variety of metrics. One issue raised by reviewer k3Ee was the choice of baseline models.

Reviewer 6bM6, for whom the paper was marginally above threshold, agreed that the paper was well-written and the dataset a useful resource, but wasn't completely convinced that the comparison between models was fair as results would also depend on the training data, and wondered to which extent the results would change with newer models.

Finally, reviewer bnQw, who ranked the paper as marginally below threshold, shared the concerns above, but also suggested that the authors should  concentrate on adult performance as that's what LLMs are meant to capture.

Summary of rebuttal:

There was no substantive criticism of the work by the reviewers, so no need for extensive rebuttals, but the concerns were all addressed.

Summary of strengths:

a. The paper is very well-written.
b. The dataset is well-designed and likely to be useful.

Summary of weaknesses:

a. It's not clear how long-lived the results of the analysis are going to be.

Summary opinion:

On balance, I would be inclined to accept this paper - the dataset is well-designed and will be a useful resource as there aren't many other datasets of this typee. And the paper is very well-written, which should increase the chances of the dataset being adopted, as one of the reviewers points out.